# Decreasing Incidence of Gastric Cancer with Increasing Time after *Helicobacter pylori* Treatment: A Nationwide Population-Based Cohort Study

**DOI:** 10.3390/antibiotics11081052

**Published:** 2022-08-03

**Authors:** Taewan Kim, Seung In Seo, Kyung Joo Lee, Chan Hyuk Park, Tae Jun Kim, Jinseob Kim, Woon Geon Shin

**Affiliations:** 1Department of Internal Medicine, Kangdong Sacred Heart Hospital, Hallym University College of Medicine, Seoul 05355, Korea; attack836@naver.com; 2Institute for Liver and Digestive Diseases, Hallym University, Chuncheon 24252, Korea; 3University Industry Foundation, Hallym University, Chuncheon 24252, Korea; lee30553969@gmail.com; 4Department of Internal Medicine, Hanyang University Guri Hospital, College of Medicine, Hanyang University, Guri 11923, Korea; yesable7@gmail.com; 5Department of Medicine, Samsung Medical Center, School of Medicine, Sungkyunkwan University, Seoul 06351, Korea; taejunk91@gmail.com; 6Department of Epidemiology, School of Public Health, Seoul National University, Seoul 03080, Korea; jinseob2kim@gmail.com

**Keywords:** *Helicobacter pylori*, gastric cancer, general population, common data model

## Abstract

Background: Treatment of *Helicobacter pylori* (HP) has been shown to reduce the risk of gastric cancer (GC) development. However, previous studies have focused on patients at high risk of GC. This study aimed to assess the effect of HP treatment on the incidence of GC in the general population. Materials and Methods: Medical records were obtained from the Common Data Model-converted sample Cohort of the National Health Insurance Service of Korea (NHIS-CDM). The target cohort included those who had been prescribed HP treatment and the comparator cohort included those who had not. The association between HP treatment and the risk of GC development was assessed using the Cox proportional hazard model. The incidences of GC according to the period after HP treatment in different age groups were analyzed using proportional trend tests. Results: After large-scale 1:4 propensity score matching, 2735 and 5328 individuals were included in the target and comparator cohorts, respectively. During the median follow-up of 6.5 years, the GC incidence was lower in the HP treatment cohort than in the comparator cohort, but this was statistically insignificant (hazard ratio [HR]: 0.76; 95% confidence interval [CI]: 0.50–1.13; *p*-value = 0.19). This trend was also observed among the older age (≥65 years, HR: 0.87; 95% CI: 0.44–1.68; *p*-value = 0.69) and male cohorts (HR: 0.82; 95% CI: 0.51–1.27; *p*-value = 0.38). Among 58,684 individuals who were treated for HP from the whole NHIS-CDM cohort, the incidence of GC consistently decreased over time and showed a marked decrease with increasing age (*p* for trend < 0.05). Conclusions: In all age groups of the general population, HP treatment could be recommended to reduce the risk of GC.

## 1. Introduction

Gastric cancer (GC) is the fifth most frequently occurring cancer and the fourth leading cause of cancer-related deaths. There were an estimated >1 million new cases of GC and 769,000 associated deaths worldwide in 2020, according to the GLOBOCAN project of the International Agency for Research on Cancer (IARC) [1]. *Helicobacter pylori* (HP), which has various virulence factors, induces local and systemic immune responses and is associated with various digestive and extra-digestive diseases [2,3]. Chronic inflammation of gastric mucosa induced by HP was revealed as an important risk factor for gastric carcinogenesis [4], and the IARC classified HP as a type I carcinogen of GC in 1994 [5]. With 810,000 attributable cases of cancer in 2018, HP was the most important infectious cause of cancer worldwide [6]. Approximately 50% of the global population is infected with HP, and the prevalence of HP and GC is especially high in East Asia, including Korea, Japan, and China [1,7].

Eradication of HP can prevent gastric carcinogenesis through the healing of mucosal inflammation and the cessation of subsequent genetic damage to gastric epithelial cells [8]. Several studies have reported that HP treatment reduced the incidence of GC, but these studies mainly included patients with early GC or pre-neoplastic lesions, or patients who received endoscopic treatment for these lesions [9,10,11]. Studies on asymptomatic and healthy HP-infected patients were also limited to relatively young patients under the age of 65 or patients with a family history of GC [12,13,14,15], so it is difficult to say that the studies targeted the general population. In several systematic reviews and meta-analyses, it was reported that HP treatment lowered the incidence of GC and mortality associated with GC. However, these studies still have limitations in that they mainly targeted the aforementioned high-risk populations of GC [16,17,18,19].

A big-data study based on the Hong-Kong territory-wide healthcare database for the effect of HP treatment on the incidence of GC was reported in 2018. In this study, HP treatment reduced the incidence of GC in individuals aged >60 years, although the control group was not HP-infected and non-treated individuals [20]. Kumar et al. [21] reported that confirmed HP eradication was associated with a lower risk of GC development in a large cohort of veterans using data from the Health Administration Corporate Data Warehouse in the United States. However, these results may be inaccurate because of the unmeasured confounders associated with GC and immortal-time, indication, and/or protopathic biases, owing to the retrospective study design without well-matched controls.

Recently, the incidence of GC worldwide showed a decreasing trend, but in younger people under 50 years of age showed an increasing trend [22]. In Korea, this tendency was also seen even in 30 to 39-year-old adults who are not subject to the national cancer screening program [23]. In addition, many previous studies have reported the effect of the HP treatment on the GC prevention, mainly in participants under the age of 65. Thus, the evidence of this effect in older age is relatively insufficient.

Therefore, we assessed the effect of HP treatment on the risk of GC in the general population. We conducted a well-designed cohort study that included a negative control, a new-user model, large-scale propensity score matching, stratification, and sensitivity analyses based on a validated nationwide dataset.

## 2. Methods

### 2.1. Data Source

In 2002, the National Health Insurance Service of Korea (NHIS) set up a representative stratified cohort of the Korean population for research purposes. This National Sample Cohort (NSC) randomly sampled 1,025,340 participants, which was 2.2% of the total eligible Korean population. The database included data on individual demographic profiles, health insurance claims, death registry, disability registry, and national health check-ups in 2002, with a follow-up period of 11 years [24]. In 2017, the NHIS-NSC data were converted into the Observational Medical Outcomes Partnership Common Data Model (OMOP-CDM) [25]. This CDM-converted version of NHIS-NSC (NHIS-CDM) allowed distributed network research and has been validated in international and national studies [26,27,28].

### 2.2. Study Design

We designed a retrospective cohort study using ATLAS, a free, public, web-based tool provided by the Observational Health Data Science and Informatics (OHDSI). The OHDSI is a global open-science collaborative for large-scale observational research using the OMOP-CDM [29]. First, we adopted a new-user model to minimize immortal time bias by at least a 1-year observatory period before entering the study. The date of prescription of antibiotics to treat HP was considered the index date. Second, new-onset GC was defined as diagnosis at least 365 days (lag period) after the index date. Third, we adjusted various risk factors for GC by extensive propensity score matching to minimize confounding bias.

The protocol of the current study was approved by the Institutional Review Board of Kangdong Sacred Heart Hospital (IRB number: 2018-05-013). The requirement for informed consent was waived because of the retrospective study design.

### 2.3. Study Populations and Cohort Definitions

The target and comparator cohorts were obtained from the NHIS-CDM based on HP treatment prescription. The included individuals were: (1) ≥40 years of age, (2) who had >1 year of observation period before the index date, and (3) who had undergone upper endoscopy within a year before the index date. The target cohort consisted of those who were prescribed the medications simultaneously for HP treatment on the same day. We included the HP treatment regimens of clarithromycin or metronidazole-based triple (clarithromycin or metronidazole, amoxicillin, and proton pump inhibitor) or bismuth-based quadruple therapy (bismuth, metronidazole, tetracycline, and proton pump inhibitor), for 7–14 days. Based on the consensus of the Korean HP study group reported in 1998 [30], metronidazole-based triple therapy was prescribed as a first-line treatment until the regimen was excluded from a new guideline in 2009 [31]. The comparator cohort comprised individuals who were not prescribed any drug included in the HP treatments concomitantly. The list of concept identifications that were used to find target drugs is provided in Appendix A.

The exclusion criteria for both cohorts were: (1) Diagnosis of GC at <1 year of cohort entry or (2) undergoing gastrectomy before cohort entry for any reason. Individuals in both cohorts were censored at the diagnosis of GC or at the end of the observation period.

### 2.4. Outcomes

The primary outcome was the comparison of GC risk between the target and comparator cohort. The diagnosis of GC was identified using the International Classification of Diseases (ICD)-10 codes of C16.0–C16.9, and D002 (carcinoma in situ of stomach). Assessment of the incidence of GC using only the ICD-10 codes is accurate as malignant diseases are enforced by law to be registered in the national cancer registration program in Korea. The overview of study design and cohort construction is shown in Appendix A.

The secondary outcome was to compare the incidence rate of GC by the presence/absence of HP treatment in the older age (≥65 years old) and male groups. Both these groups are considered to be at high risk of GC.

Additionally, we investigated the incidence of GC according to the elapsed period after HP treatment in another cohort made up of all those who were treated for HP from NHIS-CDM.

### 2.5. Statistical Analysis

The covariates used for extensive propensity score matching between the target and comparator cohorts were age, sex, index year, all recorded comorbidities, prescribed drugs in the 365 days before the index date, smoking history, alcohol consumption, body weight, family history, and Charlson comorbidity index. Regularized logistic regression models for large-scale propensity score matching were provided by the OHDSI. Propensity score matching was done in a 1:4 ratio and a caliper of 0.2 on the logit scale. The propensity score was estimated using logistic regression models with the L1 penalty hyper-parameter selected through 10-fold cross-validation using high-performance computing [32].

Hazard ratios (HRs) with 95% confidence intervals (CIs) for GC risk between the target and comparator cohorts were calculated by Cox proportional hazard models using the CohortMethod package in R. The cumulative incidence rate of GC was assessed by the Kaplan–Meier method. The incidence rates per 1000 person-years were also calculated. The incidences of GC after HP treatment by age group were compared using a proportional trend test. Two-sided *p*-values <0.05 were considered statistically significant in all comparisons. All analyses were performed using ATLAS ver. 2.7.2 and R statistical software (version 3.6.1 for Windows; R Foundation for Statistical Computing, Vienna, Austria).

### 2.6. Sensitivity and Negative Control Analyses

To confirm the reproducibility of our results, we conducted a sensitivity analysis based on various sets of propensity score matching in target-comparator ratios (1:4 and 1:1), lag periods (1 and 2 years), and propensity score stratification. We performed empirical calibration of the *p*-values by fitting an empirical null distribution to point estimates of the negative control outcomes, which were assumed to not be associated with the target or comparator cohorts [33]. In these negative control outcomes, we assumed that the true relative risk between the target and comparator cohorts was 1. Ninety selected negative control outcomes are listed in Appendix A.

## 3. Results

### 3.1. Study Flow and Baseline Characteristics

Initially, a total of 53,443 individuals were included in this study from the 1,025,340 people registered in the NHIS-NSC. Next, 21,801 and 31,642 individuals were assigned to the target and comparator cohorts, respectively. Finally, 2735 and 5328 individuals were enrolled in the target and comparator cohorts, respectively, after the exclusion of those who were included in both cohorts, who did not meet the inclusion criteria, and who were not matched in the propensity score matching model (Figure 1).

In propensity score matching, a total of 10,899 covariates were included and the maximum standardized mean difference after propensity score matching was 0.07 (Appendix A). We used funnel plots to estimate systematic errors and as the plots were symmetrical, most negative control outcomes did not differ significantly between the target and comparator cohorts (Appendix A).

The baseline characteristics of enrolled individuals in the target and comparator cohorts are shown in Table 1. Among all the covariates, we reported covariates that accounted for >5% of all participants after propensity score matching. The most common medical conditions were acute respiratory disease, hypertensive disorder, and visual system disorder. The most commonly prescribed medications before cohort entry were drugs for acid-related disorders, psycholeptics, and anti-inflammatory and anti-rheumatic products.

### 3.2. Effect of HP Treatment on GC Risk in the General Population

The median observation period was 6.6 years in the HP treatment and 6.7 years in the non-HP treatment cohort. The median intervals from cohort entry to GC diagnosis were 5.4 and 5.1 years, respectively. The Kaplan–Meier curves for cumulative incidences of GC in the target and comparator cohort are presented in Figure 2. During the observation period, 42 and 104 cases of GC had been diagnosed in the target and comparator cohorts, respectively (HP treatment vs. non-HP treatment, 42/17,938 person-years vs. 104/35,564 person-years). Although it was not statistically significant, the HP treatment cohort had a lower risk of GC than did the non-HP treatment cohort (HR: 0.76, 95% CI: 0.50–1.13, *p*-value = 0.19) (Figure 2a). We performed sensitivity analyses with different ratios of propensity score matching, adjusting the lag periods and propensity score stratification. In these analyses, the GC incidences were consistently lower in the HP treatment than in the non-HP treatment cohorts (1:4 matching with 2 year lag period, HR: 0.79, 95% CI: 0.51–1.21; 1:1 matching with 1 year lag period, HR: 0.69, 95% CI: 0.43–1.08; 1:1 matching with 2 year lag period, HR: 0.82, 95% CI: 0.49–1.36; propensity score stratification with 1 year lag period, HR: 0.78, 95% CI: 0.56–1.06; propensity score stratification with 2 year lag period, HR: 0.79, 95% CI: 0.56–1.11) (Appendix A).

### 3.3. Effect of HP Treatment on GC Risk in the High-Risk Groups (Age ≥ 65 Years and Male Sex)

In the same setting of large-scale propensity score matching with ratio 1:4 and 1 year lag period, HP treatment in individuals aged ≥65 years and men showed a decreased risk of GC development (≥65 years of age, HR: 0.87, 95% CI: 0.44–1.68, *p*-value = 0.69; male sex, HR 0.82, 95% CI: 0.51–1.27, *p*-value = 0.38) (Table 2). In addition, in the setting of different matching ratios and extended lag periods, HP treatment in individuals aged ≥65 years and men showed a trend of decreased risk of GC incidence (Appendix A). The cumulative incidence of GC in each high-risk group is shown in Figure 2b,c.

### 3.4. Incidence of GC According to the Period after HP Treatment by Age Group

Using the data of NHIS-NSC, we analyzed the incidence of GC among individuals who were treated for HP according to the period after HP treatment and age. Among the HP treatment recipients, we created five groups (18–40, 41–50, 51–60, 61–70, and 71–80 years of age), and the incidence of GC was reported every 3 years from 1–10 years after HP treatment. A total of 58,684 persons received HP treatment, and 385 cases of GC were diagnosed 1–10 years after HP treatment.

The incidences of GC by time after HP treatment across age groups are listed in Table 3. The graphs representing these results are depicted in Figure 3. We consistently found that the incidence of GC decreased with an increasing time period after HP treatment and there was a more marked reduction of GC with increasing age.

## 4. Discussion

This study was a nationwide population-based cohort that used large-scale propensity score matching models to estimate the effect of HP treatment on the risk of GC development in the general population of Korea.

Although the incidence and cancer-related mortality of GC have declined significantly in the past five decades, GC is still one of the most commonly diagnosed cancer and has high cancer-related death worldwide [34]. This decreasing trend is thought to have been contributed by proper screening tests, which are currently being implemented or being developed [35], as well as risk factor modification such as refrigeration of food, improved hygiene, and HP eradication. It has been reported that the Korean National Cancer Screening Program not only reduced the incidence of GC, but also improved the long-term survival rate of GC patients [36]. Recently, long-term use of specific medications or persistent obesity have shown the association with the risk of GC, however, one of the most important modifiable risk factors is HP [28,37,38]. The association of HP and GC was further supported by a study that showed the prevalence of HP infection has declined, which correlates with the declining trends of age-standardized incidence and mortality of GC from 1990 to 2020 in Taiwan [39].

As Korea is one of the countries with the highest prevalence of HP infection and GC, it is crucial to assess the effect of HP treatment on the risk of GC. Therefore, several studies have been conducted in Korea to try and address this. The conclusions were that HP treatment reduced the incidence of metachronous GC among those who were treated with endoscopic resection in early-stage GC [9,10] and newly developed GC among the asymptomatic healthy participants with a family history of GC [15]. However, since the personal and familial histories of GC are strong risk factors for GC, it is unclear whether the individuals studied were representative of the general population.

In this study, only a decreasing trend, although statistically insignificant, was observed in the incidence of GC among individuals who had received HP treatment, compared to those who had not (HR: 0.76, 95% CI: 0.50–1.13, *p*-value = 0.19). This effect of HP treatment on GC risk reduction was also seen in the high-risk groups of older age (≥65 years) and male sex. This could be partly explained by the proportion of individuals who were HP negative in the comparator cohort or those for whom HP eradication failed in the treatment cohort. Interestingly, the incidence of GC decreased with time after HP treatment and this reduction was greater with increasing age. Considering this delayed and profound effect with older age on reducing GC development, HP eradication should be recommended regardless of age.

A cohort study based on a large healthcare database was published in 2018, which analyzed the association between HP treatment and GC incidence in Hong Kong [20]. The incidence of GC was lower in the age and sex-matched local population than expected (*p* = 0.06). The incidence of GC was lower in individuals aged ≥60 years (standardized incidence ratio: 0.82, 95% CI 0.69–0.97) and in cases wherein ≥10 years had elapsed following HP treatment, in agreement with our results. In addition, there was a big-data study involving 370,000 veterans from the Health Administration of the United States, which showed decreased GC incidence among individuals with confirmed HP eradication. Despite the large sample size, 92.3% of the included patients were male and the mean age was 62 years [21]. In Korea, previous large-scale cohort studies found that HP treatment reduced long-term mortality but not the incidence of GC in patients with diabetes and hypertension [40,41].

The study has several strengths. First, we investigated the general population included in the Korean nationwide population-based data, which was converted into OMOP-CDM, was used. Several previous randomized controlled trials, cohort studies, systemic reviews, and meta-analyses included high-risk populations only, including individuals who had undergone endoscopic resection of GC. Hence, their findings were not applicable to the general population. Second, to reduce the effect of confounding factors, a large-scale propensity score matching was performed using 10,899 covariates. Since our database was converted into the OMOP-CDM, all medications, diagnoses, and disease severity (Charlson Comorbidity Index) were included as covariates. Recently, nationwide population-based studies on the relation between drug exposure and GC incidence were published in Korea. As GC was found to be associated with the use of proton-pump inhibitors, aspirin, metformin, and statins [28,37], we matched all of these compounds. Through the large-scale propensity score matching, some inevitable loss of cases occurred, and the statistical power was lowered. However, the treatment and comparator cohorts remained comparable. Third, we performed multiple sensitivity analyses using different matching ratios and lag periods. The analyses showed the robustness of our results. Fourth, the study was conducted in Korea, one of the countries with the highest prevalence of GC and HP in the world. Hence, this study showed the impact of HP treatment on GC in a high-risk region.

The study also had some limitations. First, HP eradication results could not be identified in the NHIS-NSC database. The intention-to-treat eradication rates of clarithromycin-based triple therapy without clarithromycin resistance test in Korea were 71.6% and 78.1% at 7 and 14 days, respectively, and that of bismuth-based quadruple therapy was 84.5% [42]. Thus, the overall eradication failure rates after the first, second, and/or third therapy were expected to be <5%, but the eradication results could not be estimated directly. In addition, early treatment of HP results in better prognoses in both benign and malignant lesions [43,44], and most patients who had confirmed HP infection might have been treated promptly. Therefore, we enrolled individuals who had undergone upper endoscopy within a year before HP treatment. Second, we could not identify the result of HP infection, such as rapid urease test or biopsy results, therefore, those who have HP infection but are not treated, might be included in the comparator group. Third, underlying endoscopic conditions, such as atrophy or metaplasia, or GC stage were not evaluated in this study. Thus, the risk factors of GC were not fully identified. Fourth, the residual biases may remain as this was an observational study. However, our results are reliable because we made every effort to reduce biases through the use of: (1) a new-user model for control immortal time bias, (2) lag periods for protopathic bias, (3) additional negative control, large scale propensity score matching, stratification, and sensitivity analyses for reducing various confounders based on the nationwide and validated big data.

## 5. Conclusions

HP treatment in the general population showed a trend in reducing the risk of GC incidence. The trend was consistent in the high-risk groups of individuals aged >65 years and men. In addition, HP treatment reduced the risk of GC in older individuals and the effect was greater with time after HP treatment. Therefore, HP treatment could be widely considered to reduce the risk of GC in the general population across age groups.

## Figures and Tables

**Figure 1 antibiotics-11-01052-f001:**
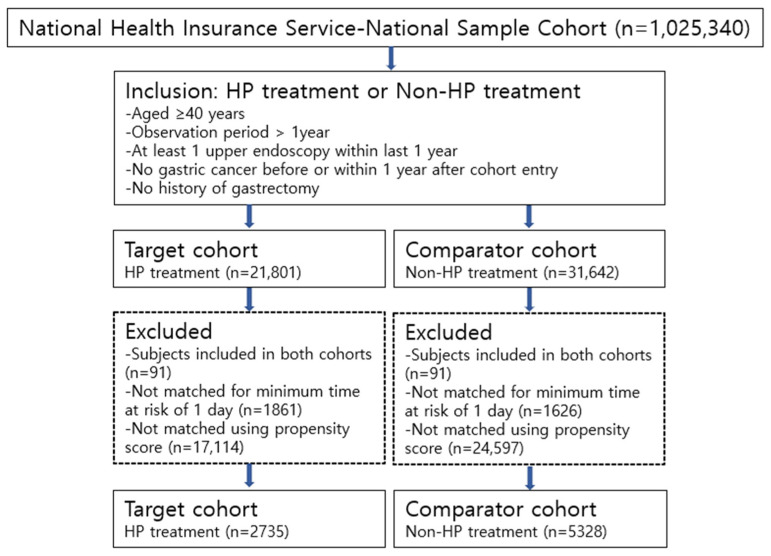
Flow chart of subject enrollment in both *Helicobacter pylori* treatment and non-*Helicobacter pylori* treatment cohorts.

**Figure 2 antibiotics-11-01052-f002:**
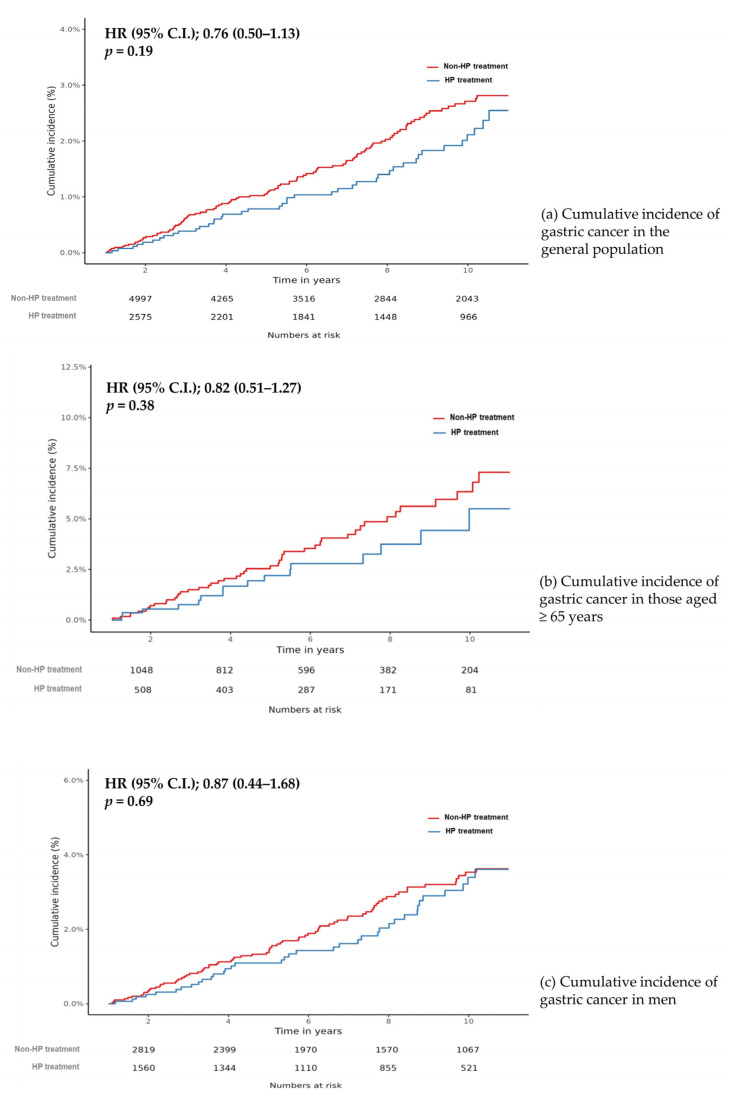
Cumulative incidence of gastric cancer in (**a**) *Helicobacter pylori* treatment cohort and non-*Helicobacter pylori* treatment cohort in the general population, (**b**) in individuals aged ≥65 years, and (**c**) in men only cohort.

**Figure 3 antibiotics-11-01052-f003:**
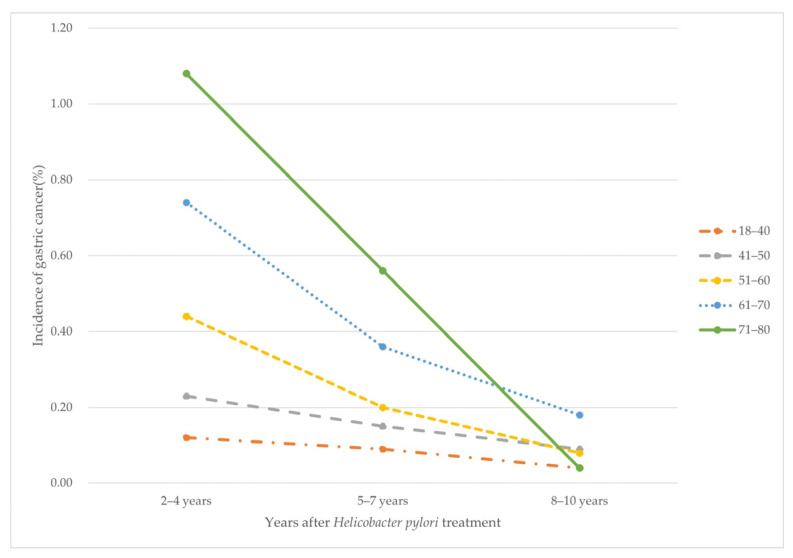
Incidence of gastric cancer by time after *Helicobacter pylori* treatment by age groups.

**Table 1 antibiotics-11-01052-t001:** Baseline characteristics of the *Helicobacter pylori* treatment and comparator groups in the general population *.

Characteristic, %	Before PS Adjustment			After PS Adjustment		
	HP Treatment (*n* = 21,801)	Non-Treatment (*n* = 31,642)	SMD	HP Treatment (*n* = 2735)	Non-Treatment (*n* = 5328)	SMD
Age group (years)						
40–44	17.9	40.6	0.52	39.2	40.2	0.02
45–49	19.8	13.4	0.17	17.6	17.7	0
50–54	18.4	10.7	0.22	13.4	13.6	0.01
55–59	14.6	9.8	0.15	10.7	10.3	0.01
60–64	11.4	10.1	0.04	8.0	8.2	0
65–69	9.2	7.7	0.05	6.3	6.0	0.01
70–74	5.3	4.5	0.04	2.8	2.5	0.02
75–79	2.5	2.0	0.03	1.1	1.0	0.02
80–84	0.8	0.8	0	0.6	0.4	0.03
85–89	0.2	0.2	0.01	<0.2	0.1	0
Sex: Female	44.0	57.0	0.26	42.6	42.4	0
Smoking	8.4	5.3	0.12	7.6	7.6	0
Alcohol consumption	32.3	27.6	0.10	27.6	27.1	0.01
Medical history						
Acute respiratory disease	60.6	62.1	0.03	47.5	47.6	0
Chronic liver disease	11.9	15.2	0.1	13.8	14.3	0.01
Depressive disorder	7.7	7.9	0.01	5.5	5.4	0
Diabetes mellitus	16.7	13.1	0.1	10.0	10.4	0.01
Gastroesophageal reflux disease	19.1	16.1	0.08	14.7	15.4	0.02
Gastrointestinal hemorrhage	13.6	8.0	0.18	10.7	10.6	0
Hyperlipidemia	30.2	17.9	0.29	18.1	19.2	0.03
Hypertensive disorder	29.0	19.6	0.22	14.3	14.5	0
Osteoarthritis	12.6	10.0	0.08	6.0	6.3	0.01
Visual system disorder	34.1	29.8	0.09	23.2	22.4	0.02
Heart disease	17.3	16.2	0.03	12.2	13.2	0.03
Ischemic heart disease	9.7	7.9	0.06	5.9	6.4	0.02
Peripheral vascular disease	10.4	5.1	0.2	4.1	4.0	0
Malignant neoplastic disease	5.6	5.6	0	3.6	3.8	0.01
Medication use						
Agents acting on the renin-angiotensin system	14.4	7.4	0.23	5.2	5.6	0.02
Antibacterials for systemic use	63.2	65.5	0.05	51.9	50.5	0.03
Antidepressants	10.8	10.9	0	6.7	6.5	0.01
Antiepileptics	5.2	3.3	0.1	2.3	2.4	0
Anti-inflammatory and antirheumatic products	58.5	56.4	0.04	43.6	43.3	0.01
Antithrombotic agents	49.7	40.1	0.19	32.1	31.6	0.01
Aspirin	14.7	8.2	0.21	6.1	6.3	−0.01
Beta blocking agents	12.2	10.4	0.06	6.9	7.2	0.01
Calcium channel blockers	18.1	11.3	0.19	7.6	9.0	0.05
Diuretics	16.3	10.6	0.17	6.9	7.3	0.02
Drugs for acid-related disorders	78.7	83.3	0.12	75.5	75.0	0.01
Drugs for obstructive airway diseases	38.6	37.2	0.03	27.1	26.9	0
Drugs used in diabetes	9.3	5.9	0.13	4.8	4.6	0.01
Metformin	6.5	3.2	0.15	2.7	2.5	0.02
Lipid modifying agent	15.2	6.2	0.3	5.3	6.1	0.03
Simvastatin	5.1	1.4	0.21	1.6	1.7	−0.01
Rosuvastatin	1.1	0.1	0.13	0.3	0.2	0.01
Pravastatin	0.9	0.7	0.03	0.4	0.8	−0.05
Pitavastatin	0.7	0.1	0.10	0.1	0.2	−0.04
Lovastatin	0.9	1.8	−0.08	1.1	1.0	0.01
Fluvastatin	0.5	0.2	0.05	0.3	0.1	0.04
Atorvastatin	6.5	1.6	0.25	1.5	1.7	−0.01
Opioids	47.5	41.8	0.11	31.4	31.9	0.01
Psycholeptics	69.6	72.1	0.06	62.5	62.4	0
Charlson index—Romano adaptation	2.3	2.0	0.03	1.3	1.1	0.02

* The applicated raw database was the National Health Insurance Service-Common Data Model (NHIS-CDM). Values are presented as the proportion of the patients (%). Abbreviation: PS, propensity matching; HP, *Helicobacter pylori*; SMD, standard mean difference.

**Table 2 antibiotics-11-01052-t002:** The effect of *Helicobacter pylori* treatment on gastric cancer incidence in the general and high-risk populations (individuals aged ≥65 years and men).

Cohorts	Number of Subjects	Observation, Person-Years	Incidence Rate of Gastric Cancer ^a^	HR (95% CI)	*p*-Value
***Helicobacter pylori* treatment in the general population**					
Non-HP treatment	5328	35,654	2.92	Ref	
HP treatment	2735	17,938	2.34	0.76 (0.50–1.13)	0.19
***Helicobacter pylori* treatment in individuals aged ≥65 years**					
Non-HP treatment	1160	6237	7.37	Ref	
HP treatment	559	2940	5.44	0.87 (0.44–1.68)	0.69
***Helicobacter pylori* treatment in men**					
Non-HP treatment	3002	19,814	3.84	Ref	
HP treatment	1652	10,713	3.45	0.82 (0.51–1.27)	0.38

^a^ Incidence rate expressed per 1000 person-years. Abbreviations: HP, *Helicobacter pylori*; HR, hazard ratio; CI, confidence interval; Ref, reference.

**Table 3 antibiotics-11-01052-t003:** The incidence of gastric cancer according to the period after *Helicobacter pylori* treatment in different age groups.

	Years after HP Treatment	2–4 Years	5–7 Years	8–10 Years	*p* for Trend
Age Group (Years)					
18–40		0.12% (17/14,775)	0.09% (13/14,758)	0.04% (6/14,745)	0.025
41–50		0.23% (41/17,573)	0.15% (27/17,532)	0.09% (15/17,505)	<0.001
51–60		0.44% (68/15,441)	0.23% (36/15,373)	0.08% (13/15,337)	<0.001
61–70		0.74% (62/8385)	0.36% (30/8323)	0.18% (15/8293)	<0.001
71–80		1.08% (27/2510)	0.56% (14/2483)	0.04% (1/2469)	<0.001
*p* for trend		<0.001	<0.001	0.018	

## Data Availability

Data are available upon in reasonable request.

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
