# Peer review of "Decreasing Incidence of Gastric Cancer with Increasing Time after Helicobacter pylori Treatment: A Nationwide Population-Based Cohort Study"

_antibiotics, 2022, doi:10.3390/antibiotics11081052_

Round 1

Reviewer 1 Report

This is a well done population study analyzing the effect of H pylori treatment on the incidence of GC. I just have few questions in the methods section as below

Methods:

Study population and cohort definition: Why prescription on the same day only is categorized as H pylori treatment? Do you get the biopsy reports the same day? Why not have a time frame of 5 or 7 days?

Was test of cure after H pylori therapy routinely done?

Why body weight instead of BMI was taken?

Reviewer 2 Report

The authors Kim et al in their manuscript titled "Decreasing Incidence of Gastric Cancer with Increasing Time 2 after Helicobacter pylori Treatment: A Nationwide Population 3 Based Cohort Study" have done a case assessment to determine the effect of HP treatment and the incidence of GC in the Korean populations. The study is very interesting and has been well analyzed. However, there are some minor concerns that would need to be addressed by the authors:

1. More details need to be provided on the time duration of the antibiotics treatment as to what the regimen was.

2. Could the authors provide information if available on patients that were not on antibiotic treatment in earlier stages but then were subjected to antibiotic treatment at a later stage?

3. This would be beyond the scope of the study but do the authors know as to if there are pathological presentation of GC post antibiotic treatment or in variation of metastases. This would be great data to be supportive of say decrease in mets or tumor occurence of a milder phenotype.

Reviewer 3 Report

Dear Authors

Thank you very much for your manuscript submission. Your study is well-designed and well-represented. However, a Minor Revision is needed:

1. You have mentioned the bacterial agent of H.pylori in Introduction section with no information in this regard. It is recommended to read and add the following paper to the References section of the manuscript to have fruitful Introduction section.

Advances in diagnosis and treatment of Helicobacter pylori infection. Acta Microbiol Immunol Hung. 2017 Sep 1;64(3):273-292. doi: 10.1556/030.64.2017.008. Epub 2017 Mar 6. PMID: 28263101.

2. Moreover, you have published several interesting papers in this field which can be used in Discussion section. So, please do read and add the following papers (including yours and some others) to have fruitful Discussion section:

Aspirin, metformin, and statin use on the risk of gastric cancer: A nationwide population-based cohort study in Korea with systematic review and meta-analysis. Cancer Med. 2022 Feb;11(4):1217-1231. doi: 10.1002/cam4.4514. Epub 2021 Dec 30.

Risk of Psoriasis in Postgastrectomy Gastric Cancer Survivors: A Nationwide Population-Based Cohort Study. Ann Dermatol. 2022 Jun;34(3):191-199. doi: 10.5021/ad.2022.34.3.191. Epub 2022 May 20. PMID: 35721330; PMCID: PMC9171185.

Association between the Persistence of Obesity and the Risk of Gastric Cancer: A Nationwide Population-Based Study. Cancer Res Treat. 2022 Jan;54(1):199-207. doi: 10.4143/crt.2021.130. Epub 2021 May 4. PMID: 33940785; PMCID: PMC8756136.

Catching Up with the World: Pepsinogen Screening for Gastric Cancer in the United States. Cancer Epidemiol Biomarkers Prev. 2022 Jul 1;31(7):1257-1258. doi: 10.1158/1055-9965.EPI-22-0372. PMID: 35775231.

Effect of gastric cancer screening on long-term survival of gastric cancer patients: results of Korean national cancer screening program. J Gastroenterol. 2022 Jul;57(7):464-475. doi: 10.1007/s00535-022-01878-4. Epub 2022 May 14. PMID: 35568752.

Declining trends of prevalence of Helicobacter pylori infection and incidence of gastric cancer in Taiwan: An updated cross-sectional survey and meta-analysis. Helicobacter. 2022 Jul 18:e12914. doi: 10.1111/hel.12914. Epub ahead of print. PMID: 35848363.

Epidemiology of stomach cancer. World J Gastroenterol. 2022 Mar 28;28(12):1187-1203. doi: 10.3748/wjg.v28.i12.1187. PMID: 35431510; PMCID: PMC8968487.

3. It is recommended to revise the Conclusion section in accordance with the recommended papers.
